# Brief communication: 'Multi-hazard-to-health-outcome' (MH<sub>2</sub>O) pathways: the known, the unknown, and ten most urgent priorities.

Harriet Moore<sup>1</sup>, Qiuhua Liang<sup>2</sup>, Lee Bosher<sup>3</sup>, John Atanbori<sup>4</sup>, Mark Gussy<sup>1</sup>, Amogh Mudbhatkal<sup>1</sup>, Joe Swift<sup>5</sup>, Jaspreet Phull<sup>6</sup>, Kirsten Guy<sup>6</sup>, Lynsey Collinson<sup>7</sup>, Andy Penny<sup>7</sup>, Maria Athanassiadou<sup>8</sup>, Kaja Milczewska<sup>9</sup>, Ebenezer Amankwaa<sup>10</sup>, Lucy Kennedy<sup>11</sup>, Edward Hanna<sup>12</sup>, Gregory Sutton<sup>12</sup>, Bartholomew Hill<sup>12</sup>, Colin Hopkirk<sup>13</sup>

Correspondence to: Harriet E Moore (HaMoore@lincoln.ac.uk)

Abstract. Climate-driven hazards like heat and flooding have complex impacts on human health. Most research considers the impact of individual hazards (e.g., heatwaves) on discrete health outcomes (e.g., heatstroke). However, climate-driven hazards often precipitate additional hazards with cumulative health impacts, such as the compound effect of drought and heatwaves on the physical and mental health of farming communities. Little is known about 'multi-hazard-to-health-outcome' (MH<sub>2</sub>O) pathways. We engaged multi-sectorial and international stakeholders through the newly established MH<sub>2</sub>O Working Group and report our co-developed ten most urgent priorities for guiding research, policy and practice towards preparing our global One Community for future uncertainty.

#### 1 Introduction

20

Climate-driven hazards including unusually high and low temperatures, flooding and extreme storms, impact human health across multiple temporal and geospatial scales, and via direct and indirect mechanistic pathways (Romanello et al., 2024). Direct pathways include heatwaves causing heatstroke, while indirect pathways often involve impacts on capital and economic productivity, such as high rates of male suicide in agricultural areas exposed to prolonged drought and economic decline (Austin & Kiem, 2025). The impact of environmental hazards on human health has been of interest to the research community and policy makers for more than 50 years, with substantial leaps forward following globally significant environmental crises

<sup>&</sup>lt;sup>1</sup>Lincoln Institute for Rural and Coastal Health, University of Lincoln, Lincoln, LN6 7TS, UK

<sup>&</sup>lt;sup>2</sup>School of Architecture, Building and Civil Engineering, Loughborough University, Loughborough, LE11 3TT, UK

<sup>&</sup>lt;sup>3</sup>School of Marketing, University of Leicester, Leicester, LE1 7RH, UK, UK

<sup>&</sup>lt;sup>4</sup>School of Engineering and Physical Sciences, University of Lincoln, Lincoln, LN6 7TS, UK

<sup>&</sup>lt;sup>5</sup>Public Health Team, Environment Agency, Bristol, BS1 5AH, UK

<sup>&</sup>lt;sup>6</sup>Lincolnshire Integrated Care Board, Sleaford, NG34 8GG, UK

<sup>&</sup>lt;sup>7</sup>developmentPlus, Lincoln, LN5 8EW, UK

<sup>&</sup>lt;sup>8</sup>Met Office, Exeter, EX1 3PB, UK

<sup>15 &</sup>lt;sup>9</sup>Environmental Hazards and Emergencies, UK Health Security Agency, London, SE1 8UG, UK

<sup>&</sup>lt;sup>10</sup>Department of Geography and Resource Development, University of Ghana, Accra, G4-489-4642, Ghana

<sup>&</sup>lt;sup>11</sup>Spottitt Ltd, Harwell, OX11 0QR, UK

<sup>&</sup>lt;sup>12</sup>School of Natural Sciences, University of Lincoln, Lincoln, LN6 7TS, UK

<sup>&</sup>lt;sup>13</sup>Every-One, Lincoln, LN5 OHU, UK

40

(Nohrstedt & Parker, 2024), including floods (Nohrstedt, 2011), droughts (Gbegbelegbe et al., 2024), and wildfires (Bryant et al., 2014). Traditionally, the study of environmental hazards has been aligned with, or 'siloed' within (Bixler et al., 2021) specialist research fields; geoscientists examine earthquake and/or landslide impacts while environmental scientists consider flooding and drought. That similar 'siloing' occurs in policy-making spheres (e.g., Leiren & Jacobsen, 2018) is not surprising; planning for compound and co-occurring hazards is an emerging frontier compared to traditional disaster risk management (Ishiwatari & Hirai, 2024). However, climate change predictions indicate that altered meteorological conditions will precipitate increasingly extreme weather events involving complex, inter-related and co-occurring 'multi-hazards' (Boult et al., 2022) with consequences for human health. Thus, understanding transitions between single and multiple hazards, and 'multi-hazard-to-health-outcomes' (MH<sub>2</sub>O) is central to climate adaptation.

The importance of quantifying multi-hazard impacts, as opposed to singular hazard-impact pathways, is not a novel observation. More than 30 years ago, Lewis (1984) considered the cumulative impact of multi-hazards on societies and evaluated the effectiveness of colonial era administration for responding to earthquakes, droughts, and hurricanes impacting the island country of Antigua. Lewis (1984) observed that, "To isolate for study each of these as they occur, would be to oversimplify the inter-relationships between the after-effects of one and the occurrence and the effects of the next." (Lewis, 1984, p.190). Arguably, this reflection is even more relevant today than >30 years ago, particularly in the context of human induced climate change and co-occurring climate-driven hazards. Rural and urban areas of countries including Australia (Chapman et al., 2025) are routinely exposed to high temperatures, drought conditions, and extreme flooding in rapid succession, precipitated by inter-related meteorological conditions (Guerreiro et al., 2018). Coastal and rural communities in the UK regularly experience ground, surface, and coastal flooding resulting from tidal surges, pluvial, and fluvial processes simultaneously (Hendry et al., 2019). The complexity of these inter-relationships will intensify as a result of climate change.

Despite the early origins of multi-hazard research, most innovations in the field have occurred relatively recently. A preliminary scoping search on Web of Science identified more than 120,000 studies about environmental hazards published from 1970 onward. Of these, 1,896 referred to 'multi-hazards', with the majority (90%) published in the last decade. Studies published in the 1990s mostly consider engineering challenges associated with the compound effect of hazards like earthquakes and storms on infrastructure. More recent publications tend to focus on vulnerability and impact mapping and mitigation. Only 149 studies were identified related to 'multi-hazards and health', with the earliest (Spencer et al., 2005) examining the impacts of volcanic eruptions, and notably appearing in this journal. More than 80% of studies on 'multi-hazards and health' were published between 2019 and 2025; 34% of the studies addressed challenges associated with climate change, and most considered impacts on physical health, while only two of the studies explored 'multi-hazard-to-health pathways' for psychiatric and psychological health. Given the paucity of research investigating 'multi-hazards-to-health-outcomes' (MH<sub>2</sub>Os), this emerging frontier represents a significant gap in our global knowledge about the likely consequences of climate change on societies.

In February 2025, we launched the Multi-Hazard-to-Health-Outcomes (MH<sub>2</sub>O) Working Group, an international, multisectorial, and interdisciplinary community of academic researchers, policy-makers, community advocates, health service providers, funding organisations, environmental agencies, lived experience experts, and clinical practitioners. A summary of the event, including attendees and presentations is documented on our <u>website</u>. Following our inaugural meeting, we synthesized the most urgent priorities co-developed by our MH<sub>2</sub>O community. Here, we report our ten recommendations for rapidly progressing evidence generation in the field of multi-hazard-to-health-outcomes.

## 2 Multi-Hazards-to-Health-Outcomes (MH2O) Working Group

The opportunity to form the MH<sub>2</sub>O Working Group arose through consultation with climate and health stakeholders about the complex inter-relationships between meteorological conditions (e.g., temperature, wind speed, wind direction), atmospheric pollutants (e.g., greenhouse gases), secondary pollutants (e.g., ground-level ozone), and the escalation of manageable health conditions (e.g., asthma, depression) to acute situations requiring emergency medical attention. The MH<sub>2</sub>O Working Group aims to synthesise existing evidence and generate new evidence about the impact that multi-hazards have on health outcomes. Our first mission is to bring together the MH<sub>2</sub>O community to identify what is known, what is not known, and to address most urgent priorities for immediate action towards supporting most vulnerable groups and regions to prepare for increasingly extreme climatic conditions through rapidly advancing research, policy, and practice.

The initial interest, and indeed request for a collaborative space to better understand the impacts of multi-hazards on health outcomes was identified during the delivery of stakeholder engagement through a multi-region (UK, Ghana) research project about the role of methane in severe respiratory and mental health outcomes (Wellcome Trust 228267/Z/23/Z). Core members (HM, JA, EA, EH, MG) of the Methane Early-Warning Network (ME-NET) research team facilitated Innovation Labs with interdisciplinary and international stakeholder groups, including representatives of climate and health institutes, communities, and government agencies. The ME-NET research aims and methods are reported elsewhere (Moore et al., 2024). Here, we are concerned with the co-created outcomes of the meeting; ten urgent priorities for research, policy and practice, and the co-production of an international community with shared vision and purpose towards understanding MH<sub>2</sub>O pathways.

#### 3 Iterative co-production:

Paradigms of participation (Chambers, 1994), engagement, inclusion, and co-production (Jasanoff, 2004) have emerged in response to the increasing complexity of phenomena defining human history, including the impact of natural hazards on the health and wellbeing of societies. While these approaches have shaped research methodology (e.g., stakeholder engagement),

<sup>&</sup>lt;sup>1</sup> https://me-net.blogs.lincoln.ac.uk/our-events/

the paradigm of co-production also recognizes that creating new knowledge in response to challenges at the nexus of natural and social processes, involves establishing new institutions with global credibility and scientific legitimacy (Miller, 2004). Seen in this light, forming the MH<sub>2</sub>O Working Group, arranging and conducting our inaugural meeting, was an exercise in co-creating an interdisciplinary global community with representation from agencies and institutes in 11 countries across the UK, Europe, Asia and Africa as much as it was a process of co-producing a mission, purpose and most urgent priorities.

The inaugural meeting involved an introduction to the state of knowledge about MH<sub>2</sub>Os, invited presentations, and co-creative problem-solving activities. In the months preceding the event, the Working Group coordinator (HM) facilitated virtual collaborative engagements with invited presenters and panellists to co-produce the event agenda and the wider narrative connecting the perspectives, lived experiences, and most urgent challenges at the frontier of preparing for, and responding to MH<sub>2</sub>Os. Presenters included not-for-profit advocacy groups (e.g., DevelopmentPlus), NHS Trusts (e.g., the Lincolnshire Integrated Care Board NHS Trust), national agencies (e.g., Environment Agency, UKHSA, Met Office), international higher education institutes (e.g., University of Ghana) and representatives of the European private remote sensing sector (e.g., EARSC, SPOTTITT). Presentations involved sharing expert knowledge and experiences about MH2O pathways, with an agreed aim to highlight impacts on most vulnerable communities, those groups, regions and areas with the greatest multihazard exposures and least capacity to engage with health protection options, including poor social and physical service access and low socio-economic mobility. Presenters were asked to consult their wider agencies and communities about most urgent priorities for dissemination during their presentations. Following the event, members of the core MH<sub>2</sub>O research team (HM, JA, EA) synthesised the shared mission, purpose and most urgent priorities presented and discussed during the in-person event, and captured through virtual tools (e.g., polls) for on-line attendees. Of 86 attendees, 33 responded to a follow-up survey inviting Working Group members to vote on the ten most urgent priorities for MH<sub>2</sub>O research, policy and practice. Thus, coproduction informed the purpose and scope fo the MH<sub>2</sub>O Working Group, as well as the future focus of our global community towards climate and health adaptation.

## 4 Mission, purpose and ten most urgent priorities:

The MH<sub>2</sub>O Working Group aims to synthesise existing evidence and generate new evidence about the impact that multihazards have on health outcomes. Our first mission is to bring together the MH<sub>2</sub>O community to identify what is known, what is not known, and to address most urgent priorities for immediate action towards supporting most vulnerable groups and regions to prepare for increasingly extreme climatic conditions through rapidly advancing research, policy, and practice. Based on the feedback of Working Group members, our most urgent priorities are presented in Table 1 in order of importance.

**Table 1.** Top ten most urgent priorities (P) for MH<sub>2</sub>O research, policy and practice.

Conducting research mapping MH<sub>2</sub>Os in under-prepared and most vulnerable communities and regions, including rural and coastal **P1** Identifying mitigation interventions to reduce the impact of climate-driven MH<sub>2</sub>Os on health and mental health, including guidelines, **P2** housing infrastructure, and environmental modelling. **P3** Translating MH<sub>2</sub>O knowledge to accessible, trustworthy, transparent, and equitable communication, education and preparation. **P4** Forming multi-inter-and-transdisciplinary networks and collaborations, including between and within sectors. Researching the compound effects of multi-hazards such as heat and air quality on mental health, including seasonal, short-term and **P5** long-term pathways. **P6** Establishing multi-datasets, assimilation methods and analysis protocols, including understanding uncertainties. Harnessing existing, novel and emerging approaches (e.g., machine learning, high-performance computing, and big data analytics) for **P7** understanding MH<sub>2</sub>Os where possible, and developing new approaches where necessary. Centring MH<sub>2</sub>O efforts on climate change for understanding future as well as current MH<sub>2</sub>O characteristics, including currently unknown **P8** 'black swan' and possible or known 'grey swan' extreme events. Disaggregating MH<sub>2</sub>O pathways, drivers and impacts between and within regions, areas, communities and individuals to make visible **P9** disparities of exposure and vulnerability, including for infectious disease and mental health, towards achieving environmental justice. Establishing outdoor-indoor pathways and compound effects for heat and air quality, including across scales and the role of classic and P10 secondary pollutants.

These focal points represent research frontiers, some of which relate to methodological processes (e.g., P6, P7) like utilising machine learning (Ferrario et al., 2025), and others reflecting substantial research gaps about specific MH<sub>2</sub>O pathways (e.g., P5, P10) and most vulnerable groups, regions, and geographic areas (e.g., P1, P9). In our view, developing climate and health multi-datasets and analytical protocols (P6) underlies most other MH<sub>2</sub>O priority research areas. To date, progress in this area has been constrained by data scarcity of important variables, and challenges with synthesising complex data across varying geospatial and temporal scales (Massazza et al., 2022). Overcoming multi-data complexities will require multi-inter-and-transdisciplinary collaboration (P4) (Bixler et al., 2021) to ensure the meaningful integration of climate and health datasets and vulnerability metrics. All priorities are symbiotic with international agreements (e.g., Paris Agreement), frameworks (e.g., the Sendai Framework), and goals, including goals and targets supported by the 2030 Agenda for Sustainable Development (United Nations, 2015) about reducing deaths and illnesses from hazards such as air pollution (3.9), reducing communicable (3.3) and non-communicable disease (3.4), and strengthening capacity for early warning and risk reduction (3.d). The coordinated generation of novel evidence, multi-datasets, and methodologies is central to achieving these targets and supporting global efforts towards climate resilience and disaster risk reduction.

# 5 Closing remarks:

Over the past three decades, research about natural hazards and societies has evolved from a focus on singular hazard-society pathways, to the cumulative effect of inter-related and co-occurring hazards on infrastructure in built environments, and most

recently the impact of multi-hazards on health outcomes. Currently, little is understood about the cumulative and compounding impacts of multi-hazards on physical and mental health, and even less about impacts on most vulnerable groups, regions and individuals. It is likely that MH<sub>2</sub>O pathways will vary over shorter and longer timeframes, across diverse urban, rural and coastal landscapes, and between as well as within communities. Unpacking these geospatial and temporal nuances will require sophisticated methods and frameworks, and novel collaborations between physical scientists with different specialisations (e.g., meteorology, hydrology and geomorphology), physical and social scientists across broad ranging specialisations, and between researchers, policy-makers, and the communities we strive to support (Bixler et al., 2021). Importantly, the voices and lived experiences of those most vulnerable to multi-hazard exposures, and least able to embrace adaptation measures should be positioned centrally to the co-production process, from defining problems to delivering and evaluating solutions. It is probable that current siloing obscures the true vulnerability of regions and people exposed to complex co-occurring hazards. Lived experience may shed light on vulnerability 'blindspots', enabling more effective disaster risk reduction.

Writing of the multiple natural hazards impacting Antigua, Lewis' (1984) concludes, "It is a simple matter to abstract these observations but it is the inter-relationships of issues which is of predominant importance – and which a separation of studies would obscure." (p.196). We echo this contention and advocate for the urgent prioritisation of multi-sectorial, interdisciplinary, and international collaborations to rapidly advance the field of 'multi-hazard-to-health-outcomes' towards preparing our global community for an uncertain future.

## **Author Contribution:**

HM contributed to conceptualisation, data curation, formal analysis, funding acquisition, methodology, project management, writing, drafting, review and editing. QL, LB, MA, KM, EH, JP, LC, CH and KG contributed to conceptualisation, drafting, reviewing and editing. JA, MG, AM, JS, GS and BH contributed to conceptualisation and project management.

## **Competing Interest:**

The authors declare that they have no conflict of interest.

## **Acknowledgements:**

Research England (E3) funding supported the research, enabling the Lincoln Institute of Rural and Coastal Health to host the Inaugural Multi-Hazards-to-Health-Outcomes (MH<sub>2</sub>O) Working Group meeting. Preliminary steps towards forming the Working Group, and follow-up investigations after the inaugural meeting were supported by Welcome Trust funding through the Ideathon Climate and Health Award (228267/Z/23/Z), in conjunction with the delivery of the Methane Early Warning Network (ME-NET) project. Contributors to the vision described in this manuscript comprise the members of the Working

Group who attended the inaugural meeting, including representatives from the University of Lincoln, East Midlands Ambulance Service NHS Trust (EMAS), Every-One (Lincolnshire), Kenyatta University (Kenya), Science and Education Development Institute (Nigeria), and Dennis Osadebay University (Nigeria). Co-creation of the ten most urgent MH<sub>2</sub>O priorities reported here was undertaken by Working Group members who completed the follow-up survey. The views expressed are those of the author(s) and not necessarily those of the UK Health Security Agency or the Department of Health and Social Care.

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
