# Peer review of "Brief communication: 'Multi-hazard-to-health-outcome' (MH2O) pathways: the known, the unknown, and ten most urgent priorities."

_EGUsphere, 2025_

## Author Comment (AC1)

RESPONSE TO THE REVIEWERS

egusphere-2025-4062

| Reviewer 1 comment | Response |
|---|---|
| Overall comment:
I very much enjoyed reading this brief communication. It is well written and it brings forward a very important research avenue.  I strongly suggest the authors to strengthen the multi-hazard risk element of their manuscript. A lot of work is done in this field, especially in recent years; also in bridging to diseases and health impacts, which is currently less recognized in the manuscript. | Thank you, we appreciate the positive feedback.

We recognize the significant contribution of the multi-hazard risk and disease fields to the wider multi-hazard agenda. Beyond contagious disease, the field of hazards/health and multi-hazards and health is evolving distinctly. Our commentary is about health inclusive of non-communicable illness, which is why we have not emphasized contagious disease research. However, we have added some references to both risk and disease in response to specific comments below. |
| While the manuscript seems to focus on "climate-driven hazards like heat and flood" many of the examples mention earthquakes and volcanic eruptions. I found this a bit confusing as these are not climate-driven. | Good point. We have made some minor revisions to clarify that while the key focus is on climate-driven hazards, we are conscious of wider narratives around environmental hazards more generally.

Please see Lines 34, 38, 45, 61. |
| I also wondered what search terms were used. Many people working in the field of (multi-) natural hazards and disasters won't refer to them as "environmental hazards". Moreover, within the field of multi-hazards, a lot of other terminology is used (e.g. cascading hazards (see Pescaroli et al 2018), consecutive disasters (De Ruiter et al 2020), etc etc). | Thanks very much for this observation and for your suggestions. Our scoping search was not intended to be exhaustive, rather to demonstrate the paucity of multi-hazard and health research compared to singular and multi-hazard research. However, including your additional search term much better reflects the overall body of research on this important topic. We excluded 'consecutive disasters' as this term returned literature related to disasters other than those associated with natural hazards. We appreciate our search does not capture all relevant literature, however, we feel the numbers much better reflect the wider field now.

We re-run our searches to include 'multi-hazards' OR 'cascading hazards' AND 'health'.

We have amended the text to reflect our findings using this new approach (including reference to disease literature) as follows: |

| | |
|---|---|
| | *'A preliminary scoping search on Web of Science identified more than 120,000 studies about environmental hazards published from 1970 onward. Of these, 4,096 referred to 'multi-hazards', or 'cascading hazards' with the majority (85%) published in the last decade.*

*Only 674 studies were identified related to 'multi-hazards', or 'cascading hazards' and 'health', with the earliest specific mention of 'multi-hazards and health' (Spencer et al., 2005) examining the impacts of volcanic eruptions, and notably appearing in this journal. More than 80% of studies on multi-hazards and health were published between 2016 and 2025; 15% of the studies addressed challenges associated with climate change, and most considered impacts on physical health with a focus on contagious disease, while only eight of the studies explored 'multi-hazard-to-health pathways' for psychiatric and psychological health.'* |
| In line 61 (but also P2 of Table 1), I wondered whether the authors truly meant (climate) mitigation or if they actually meant adaptation and/or risk reduction? | Thanks for this pick-up. We do mean adaptation and have adjusted accordingly. From a health perspective, we talk about 'mitigation interventions'; however, in the context of the wider narrative, this boils down to climate change adaptation. |
| Part of the argument made in this paper was also made in:
<li>Mora et al. 2022</li><li>Sairam & De Ruiter (2025; also published in EGUsphere).</li> | Thank you. These suggestions focus on contagious disease which we have mentioned in our top priorities. Unfortunately, we do not have space for additional citations highlighting specific health issues (e.g., contagious disease, mental health) linked to multi-hazards beyond what we have already included. |
| Line 56 – 70: the authors could also refer to recent reports by the WHO and UNDRR and that make a similar pledge. | Thank you for this suggestion. Given the strict citation limit we have had to exclude some relevant reports, particularly given the interdisciplinary nature of the argument, and the need to include literature specific to multi-hazards and health. |
| Some sentences could use a bit more careful phrasing such as "while environmental scientists consider flooding and drought". Maybe the term (socio-)hydrologists is more accurate? | We have intentionally kept the term broad (environmental scientists) as flood and drought research can also include ecologists, geomorphologists etc, as well as hydrologists. |

| | |
|---|---|
| Line 70: the authors could consider reaching out to similar groups such as the RiskKAN working group on disasters, diseases and health (see also P4 of Table 1). | We welcome all suggestions for reaching out to other groups, thank you. We plan on holding our next MH2O event in the new year and invite the reviewers to make recommendations and/or to attend themselves! |
| Instead of Bixler et al (L. 37) (and some of the subsequent references) there are a lot of studies that support this more broadly than a study that looks at a local case in Texas... I suggest the authors reflect a bit better on the field of multi-hazard risk. Eg but by no means limited to:

o   AghaKouchak et al 2014, 2018
o   Claassen et al., 2023, 2025
o   De Ridder et al 2020
o   De Ruiter et al 2020
o   Gill & Malamud 2014, 2017
o   Kappes et al. 2012
o   Quintal et al (in discussion – egusphere)
o   Scolobig et al. 2017
o   Thieken et al. 2021
o   Ward et al 2022
o   Zscheischler et al 2017, 2018 | Thank you for these suggestions. Given the limitations on number of citations, we have made some choices for inclusion that focus on wider narratives, rather than local case studies.

We have replaced the Bixler (2021) citation at (previously) Line 37 with Aghakouchak (2020). We have also added Classen et al (2023) in relation to multi-risks (Line 43) and Ridder (2020) in relation to climate change worsening multi-hazards (Line 41).

We appreciate that the field of multi-hazard risk is well developed. The field of multi-hazards and health has developed distinctly, particularly where research intersects with clinical and medical specialisations. For this reason, we have elected to include literature on challenges specific to multi-hazards and health, recognizing that we have necessarily excluded some seminal literature on multi-hazards and multi-risks more generally. |